# Influence of Varied Dietary Cholesterol Levels on Lipid Metabolism in Hamsters

**DOI:** 10.3390/nu16152472

**Published:** 2024-07-30

**Authors:** Chung-Hsiung Huang, Hung-Sheng Hsu, Meng-Tsan Chiang

**Affiliations:** Department of Food Science, College of Life Sciences, National Taiwan Ocean University, Keelung 20224, Taiwan; huangch@mail.ntou.edu.tw (C.-H.H.); superwine312@hotmail.com (H.-S.H.)

**Keywords:** Syrian hamsters, varying dietary cholesterol levels, cholesterol excretion, hyperlipidemia, hypercholesterolemia, lipid accumulation, lipid metabolism, lipid peroxidation

## Abstract

Syrian hamsters are valuable models for studying lipid metabolism due to their sensitivity to dietary cholesterol, yet the precise impact of varying cholesterol levels has not been comprehensively assessed. This study examined the impact of varying dietary cholesterol levels on lipid metabolism in Syrian hamsters. Diets ranging from 0% to 1% cholesterol were administered to assess lipid profiles and oxidative stress markers. Key findings indicate specific cholesterol thresholds for inducing distinct lipid profiles: below 0.13% for normal lipids, 0.97% for elevated LDL-C, 0.43% for increased VLDL-C, and above 0.85% for heightened hepatic lipid accumulation. A cholesterol supplementation of 0.43% induced hypercholesterolemia without adverse liver effects or abnormal lipoprotein expression. Furthermore, cholesterol supplementation significantly increased liver weight, plasma total cholesterol, LDL-C, and VLDL-C levels while reducing the HDL-C/LDL-C ratio. Fecal cholesterol excretion increased, with stable bile acid levels. High cholesterol diets correlated with elevated plasma ALT activities, reduced hepatic lipid peroxidation, and altered leptin and CETP levels. These findings underscore Syrian hamsters as robust models for hyperlipidemia research, offering insights into experimental methodologies. The identified cholesterol thresholds facilitate precise lipid profile manipulation, enhancing the hamster’s utility in lipid metabolism studies and potentially informing clinical approaches to managing lipid disorders.

## 1. Introduction

The global rise in obesity has heightened health concerns, evidenced by a significant increase in prevalence. Alongside the surge in obesity rates, associated complications, mortality rates, and healthcare costs have also escalated [1]. Notable complications include impaired glucose tolerance, reduced insulin sensitivity, and dyslipidemia, all of which pose considerable risks for cardiovascular diseases [2]. Dietary cholesterol can accumulate in plasma and tissues such as the liver [3]. Elevated plasma cholesterol is linked to diseases such as hypercholesterolemia and atherosclerosis, and can lead to hepatic lipid accumulation [3,4]. Dietary cholesterol may influence overall cholesterol metabolism, affecting plasma cholesterol levels through mechanisms like intestinal absorption, synthesis, fecal excretion, lipoprotein receptor numbers, and lipoprotein metabolic rates [5]. Obesity significantly disrupts cholesterol homeostasis by increasing cholesterol synthesis and altering lipoprotein metabolism. Studies have shown that obesity enhances the activity of HMG-CoA reductase, the key enzyme in cholesterol biosynthesis, leading to elevated cholesterol levels [6]. Additionally, obesity impairs the clearance of low-density lipoprotein (LDL) from the bloodstream, contributing to higher circulating LDL levels and an increased risk of atherosclerosis [7]. These disruptions in cholesterol metabolism underscore the critical link between obesity and dyslipidemia [8].

In the bloodstream, only 20–30% of cholesterol is absorbed from food, with the majority synthesized by the liver. Cholesterol, existing as free cholesterol or combined with long-chain fatty acids, travels in plasma lipoproteins and can be excreted via bile as cholesterol or bile salts. Cholesterol is transported within lipoproteins, which have a protein layer that makes cholesterol hydrophilic for travel through the bloodstream. Very-low-density lipoproteins (VLDLs) and LDL contribute to arterial plaque buildup, increasing heart disease risk. VLDL primarily transports triglycerides, while LDL mainly carries cholesterol. High-density lipoproteins (HDLs) return excess cholesterol from blood vessels to the liver for breakdown or use, reducing cardiovascular disease risk [9]. Moreover, a positive correlation between plasma phospholipid and cholesterol levels has been reported [10]. In obese children, plasma leptin levels are higher and correlate with body mass index, body fat percentage, and plasma levels of total cholesterol, triglycerides, and LDL-C [11]. Activity of cholesteryl ester transfer protein (CETP), which transfers cholesterol esters and phospholipids from HDL to LDL and VLDL, correlates with total plasma cholesterol and LDL-C levels [12]. Evidence suggests that consuming a diet high in cholesterol may lead to changes in lipid peroxidation and the activities of antioxidant enzymes [13]. Glutathione, in its reduced (GSH) and oxidized (GSSG) states, protects cells from lipid peroxidation in membranes. Glutathione peroxidase (GSH-Px) uses GSH to neutralize lipid peroxides, converting GSH to GSSG and preventing oxidative damage. GSH-Px and catalase activities increase in the livers of animals on high-cholesterol diets as a defense against lipid peroxidation [14]. Lipid peroxidation, initiated by free radicals attacking fatty acids, can be monitored through malondialdehyde (MDA) levels, indicating oxidative damage within biological systems. Additionally, hepatic cholesterol levels impact glucagon sensitivity, relevant to the development of non-alcoholic fatty liver disease, which is linked to both increased liver cholesterol and glucagon resistance [15].

Excessive dietary cholesterol intake has been linked to metabolic disorders such as cardiovascular disease and non-alcoholic fatty liver disease (NAFLD) [16,17]. Understanding the impact of dietary cholesterol on growth parameters, plasma lipid profiles, and lipid metabolism is essential for developing effective health strategies. Laboratory animals, including rats, mice, rabbits, and pigs, serve as valuable models for studying the effects of dietary cholesterol. These models help elucidate the physiological and biochemical responses to different cholesterol levels in the diet [18]. Each model provides unique insights due to their distinct metabolic and physiological characteristics, offering a comprehensive understanding of cholesterol’s impact across species. Growth parameters such as body weight, organ weight, and feed efficiency are crucial indicators of overall health and development. Studies have shown that dietary cholesterol can affect these parameters in a dose-dependent manner [19,20,21]. The plasma lipid profile, including total cholesterol, LDL, HDL, and triglycerides, is a key marker of cardiovascular health. Higher dietary cholesterol intake typically elevates total cholesterol and LDL levels, increasing cardiovascular risk. Monitoring plasma lipid profiles in response to dietary interventions is crucial [22]. Lipid metabolism involves complex processes of lipid synthesis, transport, and degradation, with the liver playing a central role in regulating cholesterol homeostasis and lipid storage. Dietary cholesterol influences liver metabolism by modulating enzyme activities involved in cholesterol synthesis and degradation. Excessive intake can lead to fatty liver, characterized by lipid accumulation in hepatic cells, a precursor to severe liver diseases such as steatohepatitis and cirrhosis [3]. Oxidative stress, caused by an imbalance between reactive oxygen species production and antioxidant defenses, is critical in metabolic disorders’ pathogenesis. Dietary cholesterol has been associated with increased oxidative stress, exacerbating liver damage and cardiovascular dysfunction. Investigating antioxidant potential in response to dietary cholesterol is essential for mitigating these harmful effects. Animal studies highlight antioxidants’ role in counteracting oxidative stress induced by high cholesterol diets [23]. By utilizing various animal models, scientists can better understand cholesterol’s dose-dependent effects and develop effective dietary strategies to promote health and prevent disease. These findings have significant implications for human nutrition and public health policies.

Syrian hamsters are widely used in lipoprotein metabolism studies [24]. Unlike rats and mice, hamsters exhibit a distinctive atherogenic lipoprotein profile with a significant proportion of non-HDL lipoproteins, possess CETP, engage in receptor-mediated LDL uptake via the LDL receptor, synthesize apolipoprotein (apo) B-100 in the liver, and produce apo B-48 in the intestines [20,25]. Notably, hamsters develop hypercholesterolemia and hypertriglyceridemia when fed a cholesterol-rich diet [26,27]. They are also prone to obesity and insulin resistance on high-fat, high-carbohydrate diets [28]. These traits make hamsters ideal models for evaluating drugs affecting weight, hypertriglyceridemia, and hypercholesterolemia. So far, there is limited research on how different dietary cholesterol levels impact cholesterol metabolism and antioxidant enzyme activity of hamsters. It remains unclear which cholesterol dosage best mimics human lipid metabolism and the underlying mechanisms involved. Although numerous studies have used high-cholesterol diets to induce hyperlipidemia in hamsters, we have not found any scientific findings from a single study confirming that a single variable, namely “specific cholesterol diet dosage”, triggers specific symptoms of hyperlipidemia or obesity in hamsters. Moreover, it lacks credibility to speculate on the impact of different dietary cholesterol levels based on data obtained from different studies. This study aimed to explore how different levels of dietary cholesterol affect lipid metabolism in hamsters and evaluate their impact on obesity, lipidemia, lipid accumulation, and lipid metabolism enzymes. The optimal cholesterol dosage for dietary supplementation in hamster models was determined, providing recommendations for future research on obese or hyperlipidemic animal models.

## 2. Materials and Methods

### 2.1. Animals and Experimental Diets

All animal experiments were conducted in accordance with the guidelines of the National Research Council’s Guide for the Care and Use of Laboratory Animals and approved by the NTOU Institutional Animal Care and Use Committee (NTOU IACUC 92077). Five-week-old male Syrian hamsters were obtained from the National Laboratory Animal Center of Taiwan. For acclimatization, all hamsters were housed in disinfected stainless-steel cages maintained at a temperature of 23 ± 1 °C, humidity of 40–60%, and a light cycle of 12 h per day for one week and fed solid feed (Laboratory Rodent Diet 5001, PMI Feed, Inc., St. Louis, MO, USA). Feed and distilled water were provided ad libitum. Fifty male hamsters were randomly divided into five groups based on different cholesterol addition levels: 0%, 0.1%, 0.2%, 0.5%, and 1% cholesterol groups, with 10 hamsters in each group. The number of hamsters employed in each group was based on previous studies [21,28]. Free access to water and food was provided during the feeding period. The composition of the feed is as shown in Table 1. Corn starch (African Products Limited, Germiston, South Africa), cellulose (Sigma Co., St. Louis, MO, USA), vitamins and minerals (AIN-76, MP Biomedicals, Irvine, CA, USA), and cholesterol (Wako Co., Osaka, Japan) were all purchased from distributors. Casein (TETRO PHARMA EUROPE BV, Amsterdam, The Netherlands) was purchased from Hong Sheng Instruments (Taipei, Taiwan). Body weight and food intake were measured weekly for 8 weeks. The hamsters were anesthetized with CO_2_, and blood was collected from the abdominal aorta using heparin as an anticoagulant. The collected blood was immediately centrifuged at 1570× *g* for 20 min, and the plasma was collected. Hamsters that could not provide enough volume of plasma for all analyses were excluded. Tissues of liver, kidney, heart, and adipose were also collected and weighed for further analysis. Confounders were not controlled.

### 2.2. Determination of Plasma Lipids, Leptin, CETP, and Aminotransferase

The levels of total cholesterol, free cholesterol, triglycerides, phospholipids, leptin, CETP, alanine aminotransferase (ALT), and aspartate aminotransferase (AST) were analyzed using commercially available kits (Sigma Co., St. Louis, MO, USA; Assay Designs, Inc., Ann Arbor, MI, USA; BioVision Research Products., Mountain View, CA, USA; Audit Diagnostics, Cork, Ireland). According to the characteristics of different lipoprotein densities, the ultracentrifugation method was used to separate and analyze the levels of lipoproteins in the plasma as previously described [29]. A Hitachi CP90NX ultracentrifuge (Tokyo, Japan) was used to segregate HDL-C, LDL-C, and VLDL-C in the plasma by density gradient ultracentrifugation (194,000× *g* at 10 °C for 3 h). The HDL-C, LDL-C, and VLDL-C were then recovered, and the cholesterol and triglycerides in the various separated lipoproteins were measured using the aforementioned enzymatic methods.

### 2.3. Determination of Hepatic and Fecal Lipids

According to the method of Folch et al., liver or fecal samples were homogenized with a solution consisting of 20 times its volume of chloroform/methanol mixture (2:1; *v*/*v*) using a homogenizer (Hong Sheng, SA-50 Max. 3000, Taipei, Taiwan) [30]. After homogenization, the mixture was filtered through filter paper, and the clear filtrate was then concentrated using a vacuum concentrator (Savant, Speed Vac. SC 110, 1725 rpm, Farmingdale, NY, USA) to remove organic solvents. The residue was then adjusted to a volume of 10 mL with the chloroform/methanol mixed solution (2:1; *v*/*v*) and stored in a vial for lipid analysis. The contents of lipids in liver or fecal samples were measured using the aforementioned enzymatic methods.

### 2.4. Determination of Thiobarbituric Acid Reactive Substances (TBARS) and Hepatic GSH and GSSG

To determine lipid peroxidation, the measurement of TBARS was performed by the reaction between thiobarbituric acid (TBA) and lipid peroxide product (malondialdehyde, MDA) in the samples of plasma and liver lysate. The 1,1,3,3-tetraethoxypropane (Sigma-Aldrich, St. Louis, MO, USA) was used as a standard, and the saline was used as a blank. The solutions underwent a 45-min incubation in a boiled water bath, then were allowed to cool. Following centrifugation at 1600× *g* and 4 °C for 10 min, the resulting supernatants were incubated at room temperature for 30 min. Subsequently, MDA levels were measured using a Synergy HT microplate reader (BioTek, Winooski, VT, USA) with excitation at 515 nm and emission at 553 nm.

The methods used to determine GSH and GSSG were those outlined in the previous study [31]. Briefly, the liver samples were homogenized with 1% picric acid and then centrifuged at 10,000× *g* for 20 min. From the obtained supernatant, 5 µL was used to measure total GSH. On the other hand, 100 µL of the supernatant was mixed with 2 µL of 2-vinylpyridine and allowed to react for 60 min, after which 5 µL was used for GSSG measurement. For both samples, 5 µL was added to 0.7 mL of 0.2 mM NADPH buffer, 0.1 mL of 0.6 mM DTNB, and 195 µL of distilled water, and the mixture was incubated at a constant temperature of 30 °C for 4 min. Finally, 5 µL of 200-unit GSH reductase solution was added, and the absorbance was measured at 412 nm for 3 min. The liver GSH content was determined by comparing with standard samples.

### 2.5. Statistical Analysis

The data were analyzed using SPSS/PC v28 statistical analysis software for analysis of variance (ANOVA) to test for differences among experimental groups. If differences were found, further analysis was conducted using Duncan’s multiple range test. To analyze the correlation between cholesterol supplementation dosages, hyperlipidemia, and hepatic lipid accumulation, broken line analysis was performed using SigmaPlot 14.0 to determine the broken point.

## 3. Results

### 3.1. Impact of Varied Dietary Cholesterol Levels on Body Weight, Tissue Weights, and Plasma Lipid Levels in Hamsters

After 8 weeks of feeding hamsters with varying amounts of cholesterol, it was observed that body weight decreased as the cholesterol intake increased, despite no significant change in overall food consumption (Table 2). With higher cholesterol intake, the weights of fat, heart, and kidney tissues decreased. However, the ratio of these tissue weights to body weight remained unaffected (Table 2). In contrast, the liver weight and the ratio of liver weight to body weight increased significantly with higher cholesterol intake (Table 2).

As the proportion of cholesterol in the diet increased, the plasma concentrations of total cholesterol, free cholesterol, and phospholipids also rose (Table 3). Notably, in the hamsters fed a 1% cholesterol diet, there was a significant decrease in the plasma concentrations of triglycerides (Table 3).

In examining the changes in plasma lipoproteins, a dose-dependent increase in HDL-C concentration was observed in hamsters fed diets containing 0–0.5% cholesterol. However, there was no significant difference in HDL-C concentration between the 0.5% and 1% cholesterol diets (Table 4). Plasma LDL-C and VLDL-C concentrations increased with dietary cholesterol content (Table 4). Among all groups, the highest plasma HDL-triglyceride concentration was found in the 0.2% cholesterol diet group, the highest LDL-triglyceride concentration was observed in the 1% cholesterol diet group, and the lowest VLDL-triglyceride concentration was noted in the 1% cholesterol diet group (Table 4).

### 3.2. Impact of Varied Dietary Cholesterol Levels on Lipid Accumulaton and Excretion in Hamsters

Regardless of whether measured in the concentration per gram of liver (mg/g liver) or total liver content (mg/liver), the levels of total cholesterol, free cholesterol, triglycerides, and phospholipids in hamster livers increased with higher dietary cholesterol intake (Table 5).

In terms of the impact of cholesterol intake on lipid excretion in feces, it was observed that the cholesterol content per gram of feces was higher in the hamsters fed a 0.2% cholesterol diet compared to those fed a 0% cholesterol diet (Table 6). Fecal contents of cholesterol in hamsters fed 0.5% and 1% cholesterol diets were 5 and 14 times higher than those fed a 0% cholesterol diet, respectively (Table 6). Interestingly, varying levels of dietary cholesterol supplementation did not alter the triglyceride content per gram of feces (Table 6). Although there was a trend of increased bile acid excretion in feces, statistical significance was not reached (Table 6).

### 3.3. Impact of Varied Dietary Cholesterol Levels on Lipid Peroxidation and Oxidative Damage in Tissues of Hamsters

Following an 8-week period of feeding hamsters with diets containing varied levels of cholesterol supplementation, no significant impact on lipid peroxidation in plasma was observed (Table 7). Interestingly, the concentration of TBARS in the liver was lower in the hamsters fed 0.5% and 1% cholesterol diets, although there was no statistical difference between the other groups (Table 7). Regarding antioxidant enzymes, the hepatic GSH content was lower in the hamsters fed 0.5% and 1% cholesterol diets compared to those fed a 0% cholesterol diet (Table 7). Remarkably, all hamsters received cholesterol-containing diets exhibited significantly reduced GSSG levels in the liver compared to those fed a 0% cholesterol diet (Table 7). In parallel, there were no significant differences in nephritic TBARS content among the groups (Table 7). However, in the heart, the TBARS content was significantly higher in the hamsters fed a 1% cholesterol diet compared to those fed a 0% cholesterol diet (Table 7).

Furthermore, the plasma concentration of AST was only lower in the hamsters fed a 0.5% cholesterol diet, whereas ALT concentration increased with higher cholesterol intake (Table 8). Notably, leptin concentration decreased as cholesterol intake increased, and the plasma concentration of CETP was higher in all groups compared to the group fed a 0% cholesterol diet (Table 8).

### 3.4. Correlation between Dietary Cholesterol Levels, Hyperlipidemia, and Hepatic Lipid Accumulation

In this study, we used broken line analysis to evaluate the relationship between dietary cholesterol dosage and response variables. This method helps identify threshold effects or changes in response patterns that may not be evident in a linear model. Our rationale was to avoid missing significant non-linear relationships or thresholds, allowing for a more detailed understanding of the data, especially when the response could change at specific dosage levels. Using the dose response data, we sought to determine the content at which circulating cholesterol levels match those values outlined in the Adult Treatment Panel III (ATP III). It is expected that when the added cholesterol amount was below 0.13%, the resulting plasma cholesterol level was ideally less than 200 mg/dL. Conversely, it is expected that when the added cholesterol amount exceeded 0.23%, plasma cholesterol levels became excessively high, surpassing 240 mg/dL. At an addition of 1.24%, the plasma cholesterol level is expected to reach its peak (Figure 1a). The analysis of the relationship between VLDL-C levels in plasma and the amount of added cholesterol revealed a breakpoint at 0.43%. This indicates that when the added cholesterol amount exceeded 0.43%, the increase in plasma VLDL-C slowed down (Figure 1b). Furthermore, when cholesterol was added in the range of 0% to 1%, the concentration of LDL-C in plasma exhibited a noticeable linear increase. Specifically, when the added cholesterol amount was below 0.38%, the plasma LDL-C level remained reasonable at less than 100 mg/dL. However, when the added cholesterol amount exceeded 0.97%, the plasma LDL-C level became excessively high, exceeding 190 mg/dL (Figure 1c). In terms of the correlation between added cholesterol and hepatic cholesterol content, the breakpoint was identified at 0.36%. It is expected that, beyond this point, the hepatic cholesterol content did not continue to increase linearly (Figure 1d). The relationship between added cholesterol and hepatic triglyceride content demonstrated that as the added cholesterol increased, the hepatic triglyceride content also increased linearly (Figure 1e). Lastly, the correlation between added cholesterol and hepatic phospholipid content indicates that the upward trend in phospholipid content was slowing down as the added cholesterol increased (Figure 1f).

## 4. Discussion

So far, we have found limited studies investigating the impact of cholesterol diet dosage on plasma lipid profile or obesity. However, the experimental design of these studies involved multiple variables, such as the interactions between cholesterol diet and saturated fatty acids, (*n*-3) polyunsaturated fatty acids, fat, and fructose [19,20,21]. Therefore, it was unclear whether specific dosages of cholesterol diets can induce particular symptoms in hamsters. This study is the first to show that varying specific dosages of cholesterol diets can independently induce specific symptoms in hamsters, such as elevated LDL-C, VLDL-C, and hepatic lipid accumulation, without any additional treatment. These results are crucial for future studies using hamster models to evaluate the effects of drugs or natural substances on hyperlipidemia, as they provide important information on the appropriate cholesterol dosage in the diet. This can help reduce the number of experimental groups and animals used, adhering to the 3Rs principle in further animal experiments.

Upon administering varied cholesterol levels, it was noted that at a 1% cholesterol diet, hamsters exhibited a significant reduction in body weight compared to those fed a 0% cholesterol diet. Notably, food intake remained consistent across all diets, suggesting that the observed weight change was not linked to consumption levels. The decrease in body weight could potentially result from the supplementation of 1% cholesterol, which may induce fatty liver, consequently disrupting cholesterol metabolism. Subsequently, heightened lipase activity could facilitate fat breakdown, leading to elevated levels of fatty acids in the plasma and liver. Consequently, adipose tissue reduction ensues, culminating in weight loss. Excessive cholesterol intake also yielded observable declines in weights of adipose, heart, and kidney tissues, alongside an increased liver-to-body weight ratio. Concordantly, a prior study has corroborated cholesterol accumulation in plasma and liver tissues following dietary cholesterol supplementation [3].

In normal conditions, intestinal cholesterol absorption typically reaches around 55% [32]. However, as dietary cholesterol intake increases, absorption rates decline, affecting HMG-CoA reductase activity within enterocytes [33]. While postprandial plasma cholesterol concentrations generally remain stable, prolonged excessive cholesterol intake can disrupt both plasma and hepatic cholesterol levels [34]. In this study, elevating dietary cholesterol in hamsters correlated with increased plasma total cholesterol and triglyceride levels, consistent with prior findings indicating that dietary cholesterol supplementation raises plasma cholesterol and triglyceride concentrations [35]. Additionally, plasma phospholipid concentrations rose alongside dietary cholesterol supplementation [10]. HDL-C concentration increased with cholesterol supplementation up to 0.5%, while LDL-C and VLDL-C concentrations rose steadily. VLDL-triglyceride content initially increased with dietary cholesterol, peaking before decreasing at 1%, likely due to cholesterol accumulation in the liver replacing triglycerides in lipoprotein composition. Cholesterol supplementation can either stimulate or inhibit HDL secretion, with our experiment noting an increase in HDL-C concentration. However, the HDL-C/LDL-C ratio declined with rising cholesterol supplementation, implying an increased risk of cardiovascular disease. Excessive cholesterol intake elevated HDL-C, VLDL-C, and LDL-C concentrations to facilitate cholesterol transport back to the liver for metabolism or synthesis. Notably, dietary cholesterol supplementation significantly inhibited LDL receptor activity, impairing LDL cholesterol metabolism and elevating plasma LDL levels [36]. Moreover, fecal cholesterol excretion increased with higher cholesterol intake, leading to higher excretion of acidic and neutral sterols [37]. Concordantly, there was an observed increasing trend in bile acid excretion in hamsters with higher cholesterol intake in this study. Higher cholesterol intake leads to increased bile acid excretion to maintain cholesterol balance in the body [38].

It has been showcased that both high-fat and low-fat diets supplemented with cholesterol exhibited lower plasma TBARS concentrations compared to their non-supplemented counterparts, with particularly lower levels observed in that of the low-fat diet [39]. Cholesterol is known as a potential free radical scavenger, positing its antioxidant properties [40]. Conversely, Gokkusu et al. found higher plasma TBARS concentrations in rats on a high-cholesterol diet, indicating cholesterol may exacerbate oxygen free radical production, potentially fostering arteriosclerosis [41]. In this study, we observed lower plasma TBARS concentrations in cholesterol-supplemented groups compared to the control group when cholesterol reached or exceeded 0.5%, hinting at potential antioxidant functions in hamster plasma. As dietary cholesterol supplementation increased, observable changes emerged in the liver, suggestive of hepatomegaly or fatty liver development. While no significant difference in AST concentrations was noted among groups, ALT levels rose with escalating cholesterol supplementation, often indicative of liver dysfunction, potentially linked to fatty liver progression [42]. Hepatic cholesterol accumulation and triglyceride content also increased with dietary cholesterol supplementation, likely stemming from an imbalance between cholesterol intake and hepatic metabolism. It has been proposed that elevated hepatic triglyceride levels could result from increased triglyceride synthesis and reduced VLDL secretion, while hepatic TBARS content decreased with increasing cholesterol supplementation, suggesting that cholesterol addition may mitigate hepatic MDA formation [35,39]. Furthermore, GSH, an antioxidant, decreased with cholesterol supplementation, possibly due to cholesterol’s antioxidant properties reducing GSH demand. While no significant differences were noted in kidney TBARS among groups, heart TBARS levels increased with cholesterol supplementation. In the current study, leptin levels decreased with escalating cholesterol supplementation, potentially linked to decreased adipose tissue mass. Remarkably, following cholesterol supplementation, plasma CETP concentrations increased, correlating positively with LDL-C levels. CETP activity likely rises with LDL-C concentration, facilitating cholesterol ester transport in HDL.

Using the ATPIII guidelines as a reference, cholesterol addition below 0.13% maintained ideal plasma cholesterol lower than 200 mg/dL, and that below 0.38% keep plasma LDL-C lower than 100 mg/dL. To induce hypercholesterolemia in animal models for research, several considerations arise, e.g., elevating LDL-C concentration, increasing VLDL-C concentration, inducing hypercholesterolemia with fatty liver, and inducing hypercholesterolemia without excessive cholesterol accumulation in the liver and abnormal VLDL-C excretion. Based on the results of this study, to raise LDL-C, cholesterol can be added up to 0.97%. However, severe cholesterol accumulation in the liver and disrupted VLDL secretion may occur, possibly due to the near-saturation of liver cholesterol leading to elevated cholesterol content in VLDL. This may result from decreased LDL receptor numbers in the liver, perpetuating elevated LDL-C levels [43]. For elevating VLDL-C concentration without causing secretion abnormalities, cholesterol addition should consider the rate of VLDL-C secretion. A critical point emerged when cholesterol addition reached 0.43%, where hepatic cholesterol accumulation approached a high level while maintaining normal VLDL-C secretion. Consequently, the LDL-C concentration can reach 120 mg/dL, within the normal range according to ATPIII standards. The recommended cholesterol addition to induce hypercholesterolemia without causing abnormalities in liver function and lipoprotein secretion is 0.43%. However, considering the ATPIII LDL-C standard of 190 mg/dL may lead to liver dysfunction and fatty liver formation, adding cholesterol up to 0.97% is necessary to specifically study LDL-C changes.

## 5. Conclusions

Syrian hamsters are excellent models for lipid metabolism studies due to their sensitivity to dietary cholesterol. This study explored the effects of varying cholesterol levels on lipid metabolism and oxidative damage over an 8-week period. Based on the current results, it is expected that cholesterol supplementation below 0.13% maintains normal lipids, 0.97% raises LDL-C, 0.43% increases VLDL-C, and above 0.85% elevates liver lipids. A 0.43% supplementation is ideal for inducing hypercholesterolemia without liver damage or abnormal lipoprotein secretion. Dietary cholesterol supplementation increased liver weight and plasma levels of total cholesterol, LDL-C, and VLDL-C, as well as fecal cholesterol content, while decreasing the HDL-C/LDL-C ratio. A 1% cholesterol diet elevated plasma ALT activities, reduced hepatic TBARS values, and altered leptin, and CETP levels. These findings offer valuable guidelines for using Syrian hamsters in hyperlipidemia research, providing specific cholesterol supplementation levels to achieve desired lipid profiles while minimizing adverse effects.

## Figures and Tables

**Figure 1 nutrients-16-02472-f001:**
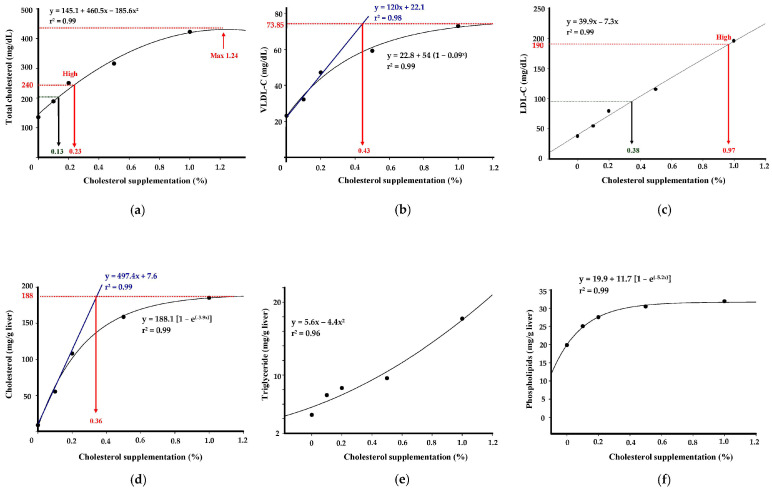
Correlation between cholesterol supplementation dosages and levels of plasma and hepatic lipids in hamsters. Hamsters were fed different experimental diets containing 0–1% cholesterol for 8 weeks, and the plasma and liver tissues were harvested to determine the levels of plasma total cholesterol (**a**), VLDL-C (**b**), LCL-C (**c**), hepatic cholesterol (**d**), triglyceride (**e**), and phospholipid (**f**).

**Table 1 nutrients-16-02472-t001:** Composition of the experimental diets.

Ingredient (%; *w*/*w*)	0% ^3^	0.1% ^3^	0.2% ^3^	0.5% ^3^	1% ^3^
Casein	20	20	20	20	20
Ching-shang oil	3	3	3	3	3
Corn oil	7	7	7	7	7
Vitamin mixture ^1^	1	1	1	1	1
Salt mixture ^2^	4	4	4	4	4
Cholesterol	0	0.1	0.2	0.5	1.0
Choline chloride	0.2	0.2	0.2	0.2	0.2
L-Cystine	0.3	0.3	0.3	0.3	0.3
Cellulose	5	5	5	5	5
Sucrose	25	25	25	25	25
Corn starch	34.5	34.4	34.3	34	33.5

^1^ AIN 76 vitamin mixture. ^2^ AIN 76 mineral mixture. ^3^ Percentage of cholesterol contained in the experimental diets.

**Table 2 nutrients-16-02472-t002:** Body weight, food intake and tissue weight in hamsters fed different experimental diets for 8 weeks.

Amounts of Cholesterol in the Diets	0%	0.1%	0.2%	0.5%	1%
Initial body weight (g)	80.7 ± 5.7	80.6 ± 6	80.5 ± 4.8	80.6 ± 4.8	80.4 ± 5.1
Final body weight (g)	107.9 ± 9.2 ^a^	105 ± 8.6 ^ab^	103 ± 6.5 ^abc^	100.1 ± 5.5 ^bc^	96.2 ± 8.9 ^c^
Food intake (g)/day	9.5 ± 1.1	9.9 ± 1	10 ± 1	11.1 ± 0.9	10.1 ± 1
Weight gain (g)	29.6 ± 11.1 ^a^	26.8 ± 10.7 ^a^	24.8 ± 6.9 ^a^	21 ± 8 ^a^	18.6 ± 8.6 ^b^
Liver weight (g)	3.9 ± 0.5 ^d^	5 ± 0.9 ^c^	5.2 ± 0.6 ^c^	6.4 ± 0.8 ^b^	7.7 ± 0.9 ^a^
Liver weight (g)/100 g B.W.	3.6 ± 0.25 ^d^	4.8 ± 1 ^c^	5 ± 0.5 ^c^	6 ± 0.9 ^b^	8 ± 1.6 ^a^
Kidney (g)	1.1 ± 0.2 ^a^	1.08 ± 0.1 ^ab^	1.05 ± 0.1 ^ab^	1.04 ± 0.07 ^ab^	0.99 ± 0.07 ^b^
Kidney (g)/100 g B.W.	1.02 ± 0.1	1.03 ± 0.06	1.02 ± 0.07	1.04 ± 0.09	1.04 ± 0.17
Heart (g)	0.4 ± 0.02 ^a^	0.39 ± 0.03 ^ab^	0.37 ± 0.02 ^ab^	0.369 ± 0.03 ^bc^	0.35 ± 0.02 ^c^
Heart (g)/100 g B.W.	0.37 ± 0.02	0.37 ± 0.02	0.369 ± 0.02	0.369 ± 0.02	0.367 ± 0.04
Adipose tissue weight (g)	2.14 ± 0.52 ^a^	1.96 ± 0.42 ^ab^	1.85 ± 0.26 ^ab^	1.76 ± 0.1 ^b^	1.6 ± 0.38 ^b^
Adipose tissue weight (g)/100 g B.W.	1.96 ± 0.33	1.85 ± 0.25	1.79 ± 0.19	1.76 ± 0.19	1.68 ± 0.46

Each value is expressed as mean ± S.D. for 8–10 hamsters per dietary group. Values in the same row with the different superscript letters are significantly different (*p* < 0.05).

**Table 3 nutrients-16-02472-t003:** Plasma levels of lipids in hamsters fed the different experimental diets for 8 weeks.

Amounts of Cholesterol in the Diets	0%	0.1%	0.2%	0.5%	1%
Total cholesterol (mg/dL)	135 ± 11.3 ^e^	189 ± 14.2 ^d^	250 ± 16.1 ^c^	316 ± 35.3 ^b^	422 ± 69.8 ^a^
Free cholesterol (mg/dL)	31.6 ± 2.43 ^e^	40.6 ± 2.95 ^d^	52.7 ± 3.49 ^c^	67.1 ± 8.69 ^b^	90.9 ± 14.6 ^a^
Cholesterol ester (mg/dL)	104 ± 9.1 ^e^	148 ± 12 ^d^	198 ± 13 ^c^	256 ± 37 ^b^	332 ± 55.7 ^a^
Phospholipid (mg/dL)	221 ± 11.9 ^d^	272 ± 15.2 ^c^	327 ± 23 ^b^	376 ± 21 ^a^	385 ± 34.7 ^a^
Triglyceride (mg/dL)	90 ± 21.2 ^ab^	87.1 ± 15.5 ^ab^	104 ± 17.2 ^a^	86.3 ± 23.6 ^ab^	74.7 ± 20.9 ^b^

Each value is expressed as mean ± S.D. for 8–10 hamsters per dietary group. Values in the same row with the different superscript letters are significantly different (*p* < 0.05).

**Table 4 nutrients-16-02472-t004:** Plasma levels of lipoprotein lipids in hamsters fed the different experimental diets for 8 weeks.

Amounts of Cholesterol in the Diets	0%	0.1%	0.2%	0.5%	1%
HDL-C (mg/dL)	74.7 ± 6.6 ^d^	101.6 ± 7.4 ^c^	124.2 ± 10.6 ^b^	142.5 ± 11.4 ^a^	148.2 ± 11.7 ^a^
LDL-C (mg/dL)	37.6 ± 3.9 ^d^	54.4 ± 6.5 ^d^	79.1 ± 11.9 ^c^	115.2 ± 23.1 ^b^	195.5 ± 37.1 ^a^
VLDL-C (mg/dL)	23.1 ± 6.3 ^d^	32.1 ± 9.8 ^cd^	47.1 ± 13.5 ^bc^	59.1 ± 19.7 ^ab^	72.8 ± 28.3 ^a^
HDL-triglyceride (mg/dL)	17.3 ± 2.7 ^b^	16.9 ± 1.9 ^b^	19.4 ± 2.6 ^a^	16.6 ± 1.7 ^b^	15.2 ± 1.9 ^b^
LDL-triglyceride (mg/dL)	6.2 ± 4.1 ^b^	5.4 ± 1.5 ^b^	5.5 ± 2.4 ^b^	7.3 ± 1.6 ^b^	12.1 ± 5.9 ^a^
VLDL-triglyceride (mg/dL)	67.5 ± 18 ^a^	64.8 ± 13.7 ^a^	79.2 ± 14 ^a^	64.6 ± 22.7 ^a^	47.3 ± 14.7 ^b^
HDL-C/LDL-C	2 ± 0.2 ^a^	1.87 ± 0.2 ^a^	1.6 ± 0.2 ^b^	1.3 ± 0.2 ^c^	0.79 ± 0.2 ^d^

Each value is expressed as mean ± S.D. for 8–10 hamsters per dietary group. Values in the same row with different superscript letters are significantly different (*p* < 0.05).

**Table 5 nutrients-16-02472-t005:** Hepatic levels of lipids in hamsters fed the different experimental diets for 8 weeks.

Amounts of Cholesterol in the Diets	0%	0.1%	0.2%	0.5%	1%
**Total cholesterol**					
(mg/g liver)	8.65 ± 3.5 ^e^	55.2 ± 11.7 ^d^	108.1 ± 16.1 ^c^	158.6 ± 14 ^b^	185.2 ± 11.6 ^a^
(mg/liver)	34.3 ± 17.1 ^e^	262 ± 59.3 ^d^	553.1 ± 72.8 ^c^	1048.3 ± 80 ^b^	1426.2 ± 178 ^a^
**Triglyceride**					
(mg/g liver)	4.5 ± 0.88 ^c^	7.2 ± 2.1 ^b^	8.2 ± 2.2 ^b^	9.6 ± 1.4 ^b^	17.7 ± 5.7 ^a^
(mg/liver)	17.8 ± 5 ^d^	35.9 ± 11 ^c^	42.9 ± 13.3 ^c^	61.2 ± 12.2 ^b^	140.5 ± 42 ^a^
**Phospholipid**					
(mg/g liver)	19.8 ± 1.2 ^c^	25 ± 2.6 ^b^	27.5 ± 2.4 ^b^	30.4 ± 4.3 ^a^	31.9 ± 3.1 ^a^
(mg/liver)	77.2 ± 9.4 ^d^	125.9 ± 32.6 ^c^	141.2 ± 14.5 ^c^	196.1 ± 41.6 ^b^	244.4 ± 39.5 ^a^
**Free cholesterol**					
(mg/g liver)	3.4 ± 0.3 ^d^	8.9 ± 1.3 ^c^	11.9 ± 2.4 ^b^	14.2 ± 4 ^b^	25.4 ± 4.1 ^a^
(mg/liver)	77.2 ± 9.4 ^d^	125.9 ± 32.6 ^c^	141.2 ± 14.5 ^c^	196.1 ± 41.6 ^b^	244.4 ± 39.5 ^a^

Each value is expressed as mean ± S.D. for 8–10 hamsters per dietary group. Values in the same row with different superscript letters are significantly different (*p* < 0.05).

**Table 6 nutrients-16-02472-t006:** Fecal levels of lipids and bile acid in hamsters fed the different experimental diets for 8 weeks.

Amounts of Cholesterol in the Diets	0%	0.1%	0.2%	0.5%	1%
Dry weight (g/day)	1.95 ± 0.27	1.76 ± 0.8	2.04 ± 0.23	1.91 ± 0.26	1.99 ± 0.41
Cholesterol (mg/g)	3.52 ± 1.2 ^d^	4.01 ± 1.1 ^cd^	6.18 ± 1.3 ^c^	15.9 ± 1.9 ^b^	48.4 ± 4.9 ^a^
Triglyceride (mg/g)	4.27 ± 2.4	5 ± 1.4	3.8 ± 1.9	4.56 ± 1.9	4.72 ± 1.4
Bile acid (μ mol/L)	83.1 ± 24	88.5 ± 24	93.9 ± 47	93.8 ± 41	94.9 ± 32

Each value is expressed as mean ± S.D. for 8–10 hamsters per dietary group. Values in the same row with different superscript letters are significantly different (*p* < 0.05).

**Table 7 nutrients-16-02472-t007:** TBARS values and antioxidant enzyme activities in hamsters fed the different experimental diets for 8 weeks.

Amounts of Cholesterol in the Diets	0%	0.1%	0.2%	0.5%	1%
**Plasma**					
TBARS (n mole/mL)	5.1 ± 0.4 ^ab^	5.1 ± 0.9 ^ab^	5.4 ± 0.7 ^a^	4.5 ± 0.6 ^b^	4.6 ± 1.0 ^b^
**Liver**					
TBARS (n mole/g)	13.8 ± 2 ^a^	9.9 ± 2.7 ^b^	10.2 ± 2.0 ^b^	6.3 ± 2.6 ^c^	6.3 ± 3.2 ^c^
GSH (n mole/mL)	24.2 ± 0.2 ^a^	24.5 ± 1.7 ^a^	24.5 ± 1.8 ^a^	21.1 ± 2.4 ^b^	17.8 ± 2.3 ^b^
GSSG (n mole/mL)	3.0 ± 0.2 ^a^	2.3 ± 0.2 ^b^	2.1 ± 0.2 ^b^	2.1 ± 0.6 ^b^	2.4 ± 0.4 ^b^
**Kidney**					
TBARS (n mole/g)	24.7 ± 5.0	24.1 ± 4.4	26.5 ± 5.6	21.5 ± 2.2	25.8 ± 7.3
**Heart**					
TBARS (n mole/g)	9.9 ± 2.7 ^b^	12.2 ± 3.5 ^ab^	17.4 ± 6.3 ^a^	12.1 ± 5.0 ^ab^	16.7 ± 7.2 ^a^

TBARS: thiobarbituric acid reactive substances; GSH: glutathione; GSSG: glutathione disulfide. Each value is expressed as mean ± S.D. for 8–10 hamsters per dietary group. Values in the same row with different superscript letters are significantly different (*p* < 0.05).

**Table 8 nutrients-16-02472-t008:** Plasma levels of transaminase, leptin, and CETP in hamsters fed the different experimental diets for 8 weeks.

Amounts of Cholesterol in the Diets	0%	0.1%	0.2%	0.5%	1%
AST (SF units)	45 ± 15 ^a^	40 ± 15 ^ab^	36 ± 7.7 ^ab^	32 ± 4.6 ^b^	47 ± 14 ^a^
ALT (SF units)	18 ± 5.7 ^c^	22 ± 13 ^c^	27 ± 10 ^c^	41 ± 13 ^b^	73 ± 18 ^a^
Leptin (pg/mL)	174.5 ± 58 ^a^	135 ± 38 ^b^	108 ± 18 ^bc^	96 ± 13 ^c^	100 ± 15 ^c^
CETP (n mole/mL)	2.1 ± 0.15 ^b^	2.54 ± 0.26 ^a^	2.49 ± 0.21 ^a^	2.6 ± 0.19 ^a^	2.6 ± 0.14 ^a^

AST: aspartate aminotransferase; ALT: alanine transaminase. Each value is expressed as mean ± S.D. for 8–10 hamsters per dietary group. Values in the same row with different superscript letters are significantly different (*p* < 0.05).

## Data Availability

The original contributions presented in the study are included in the article.

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
