# Peer review of "Influence of Varied Dietary Cholesterol Levels on Lipid Metabolism in Hamsters"

_nutrients, 2024, doi:10.3390/nu16152472_

Round 1

Reviewer 1 Report

Comments and Suggestions for Authors

The article submitted for review concerns the influence of varied dietary cholesterol levels on lipid metabolism in hamsters. The manuscript is a full article examining the effects of different levels of dietary cholesterol on obesity, the plasma lipid profile, lipid accumulation, potential damage, lipid and metabolizing enzymes. The research was also aimed at determination of the optimal cholesterol dosage for dietary supplementation in hamster models providing recommendations for future research on obese or hyperlipidemic animal models. The article is quite well written, but there are a few points that could make it better and a few points that are questionable. The proposals and main comments to the Authors are listed below.

1.       "Abstract" section should include key results and the applicability of the research conducted should be emphasized. Please provide the key (quantitative) results of the research carried out. Additionally, in the current version, the vast majority of it only concerns the experimental setup.

2.       More specific terms should be provided in the "keywords" section

3.       The "Introduction" section should expand information on research on the effect of different doses of dietary cholesterol on basic growth parameters, plasma lipid profile, lipid metabolism, including liver metabolism, fatty liver, antioxidant potential in laboratory animals (various models, e.g. rats, mice, rabbits, pigs). There is a lot of data on this subject in the literature (dozens, if not hundreds).

4.       In the “Materials and methods” section, section 2.1. "Animals and experimental diets" it is necessary to provide the consent number of the bioethics committee to conduct research involving animals. Additionally, the number of animals in each experimental group raises considerable concerns for ethical reasons. In such obvious research on the impact of cholesterol supplementation in animal diets, why conduct experiments on 10 hamsters? The results of these studies, in many different research models, have been known and widely recognized for several decades.

5.       The presented article does not bring anything new to the known and widely recognized worldwide research results on the influence of cholesterol in an atherogenic diet in various research models of experimental animals (e.g. rats, guinea pigs, mice, rabbits, dogs, pigs...). The literature describes these issues very extensively. Starting with information on determining "safe" doses of cholesterol, through "effective" doses of cholesterol in the diet (aimed to induce a hyperlipidemic effect in atherosclerotic models), ending with lethal doses of cholesterol in the diet of laboratory animals. Also well known in the literature is the effect of cholesterol supplementation (at various levels of its addition), both on growth parameters, dietary utilization rates, plasma lipid profile, lipid metaolism, degree of fatty liver, atherogenic changes in other organs (including the aorta), and the antioxidant potential of plasma and even histological parameters in various organs and in various research models (rats, mice, poultry, rabbits, etc.). Is the only novelty of this article a study on the influence of cholesterol (on the above-mentioned indicators in the body) on Syrian hamsters? Because unfortunately this article doesn't bring anything new.

6.       Taking into account animal welfare and moving away from unnecessary animal research (the results presented in this article seem to be unnecessary from a scientific point of view), it is necessary to explain what the specific purpose of this research was? What do they contribute to the broadly understood literature on this topic? They did not examine anything except the effect of cholesterol, and this has been widely known in the literature for several decades. Additionally, most studies in the literature, in addition to the effects of dietary cholesterol supplementation on the body, also present the impact of other dietary supplements, which are atherogenic. Maybe if it were part of such research, i.e. research on some other factor in the diet involving cholesterol (hypercholesterolemic model), it would not be so blatant.

7.       The presented discussion of the obtained results must ABSOLUTELY be extended to include data commonly known in the literature on this issue. Because it is ABSOLUTELY NOT TRUE that the quote from the authors of this manuscript "So far, there is limited research on how different dietary cholesterol levels impact cholesterol metabolism and antioxidant enzyme activity of hamsters. It remains unclear which cholesterol dosage best mimics human lipid metabolism and the underlying mechanisms involved” [Lines 71 to 73].

8.       There is a huge amount of data on this subject in the literature. They may not apply to hamsters, but they apply to almost all other laboratory animals. I am the author or co-author of several such works over the last 15-20 years!

9.       Therefore, It is ABSOLUTELY necessary to clarify the validity of these studies. Therefore, among others in the conclusions, the innovativeness and application possibilities of this research should be explained, emphasized and indicated. And it's not that the use of another laboratory animal model, the Syrian hamster, is new. This is nothing new considering the body's overall response to different concentrations of dietary cholesterol. This answer is either the same or very similar in various laboratory animal models.

10.   The references used are appropriate, but given the existing data in the literature, they are very sparse. Increasing the number of references would perhaps help better substantiate the relevance of the presented results in the context of the existing body of research and provide a more comprehensive review of the literature. It would also be beneficial to include more recent research and a wider range of sources to compare and contrast with current findings.

Author Response

Reviewer 1:

The article submitted for review concerns the influence of varied dietary cholesterol levels on lipid metabolism in hamsters. The manuscript is a full article examining the effects of different levels of dietary cholesterol on obesity, the plasma lipid profile, lipid accumulation, potential damage, lipid and metabolizing enzymes. The research was also aimed at determination of the optimal cholesterol dosage for dietary supplementation in hamster models providing recommendations for future research on obese or hyperlipidemic animal models. The article is quite well written, but there are a few points that could make it better and a few points that are questionable. The proposals and main comments to the Authors are listed below.

  1. "Abstract" section should include key results and the applicability of the research conducted should be emphasized. Please provide the key (quantitative) results of the research carried out. Additionally, in the current version, the vast majority of it only concerns the experimental setup.

Response:   We thank the Reviewer for the valuable suggestion. The abstract has been rewritten to include key quantitative results and the applicability of the research. (line 9-25)

  1. More specific terms should be provided in the "keywords" section

Response:   We express our gratitude to the Reviewer for the valuable suggestion. More specific terms have been provided in the "keywords" section. (line 26-27)

  1. The "Introduction" section should expand information on research on the effect of different doses of dietary cholesterol on basic growth parameters, plasma lipid profile, lipid metabolism, including liver metabolism, fatty liver, antioxidant potential in laboratory animals (various models, e.g. rats, mice, rabbits, pigs). There is a lot of data on this subject in the literature (dozens, if not hundreds).

Response:   The valuable suggestion from the Reviewer is greatly appreciated. An introduction section has been added to the revised manuscript, expanding on research regarding the effects of different doses of dietary cholesterol on basic growth parameters, plasma lipid profile, lipid metabolism, including liver metabolism, fatty liver, and antioxidant potential in laboratory animals. (line 73-103)

  1. In the “Materials and methods” section, section 2.1. "Animals and experimental diets" it is necessary to provide the consent number of the bioethics committee to conduct research involving animals. Additionally, the number of animals in each experimental group raises considerable concerns for ethical reasons. In such obvious research on the impact of cholesterol supplementation in animal diets, why conduct experiments on 10 hamsters? The results of these studies, in many different research models, have been known and widely recognized for several decades.

Response:   We are thankful to the Reviewer for the valuable input. The consent number from the bioethics committee for conducting research involving animals has been provided in the “Materials and methods” section of the revised manuscript. The number of hamsters employed in each group was based on previous studies, which commonly used 10 hamsters per group as models of hyperlipidemia. (line 129-131, 139-140)

  1. The presented article does not bring anything new to the known and widely recognized worldwide research results on the influence of cholesterol in an atherogenic diet in various research models of experimental animals (e.g. rats, guinea pigs, mice, rabbits, dogs, pigs...). The literature describes these issues very extensively. Starting with information on determining "safe" doses of cholesterol, through "effective" doses of cholesterol in the diet (aimed to induce a hyperlipidemic effect in atherosclerotic models), ending with lethal doses of cholesterol in the diet of laboratory animals. Also well known in the literature is the effect of cholesterol supplementation (at various levels of its addition), both on growth parameters, dietary utilization rates, plasma lipid profile, lipid metaolism, degree of fatty liver, atherogenic changes in other organs (including the aorta), and the antioxidant potential of plasma and even histological parameters in various organs and in various research models (rats, mice, poultry, rabbits, etc.). Is the only novelty of this article a study on the influence of cholesterol (on the above-mentioned indicators in the body) on Syrian hamsters? Because unfortunately this article doesn't bring anything new.

Response:   We thank the Reviewer for pointing out the important issue. Although hamsters are ideal experimental animals for inducing hyperlipidemia, no study has yet confirmed that a single variable, namely "specific cholesterol diet dosage," triggers specific symptoms of hyperlipidemia or obesity in hamsters. This study is the first to demonstrate that different specific dosages of cholesterol diets can induce particular symptoms in hamsters, such as elevated LDL-C, VLDL-C, and hepatic lipid accumulation. These results are crucial for future studies using hamster models to evaluate the effects of drugs or natural substances on hyperlipidemia, as they provide important information on the appropriate cholesterol dosage in the diet. This can help reduce the number of experimental groups and animals used, adhering to the 3Rs principle in animal experiments. (line 115-120, 325-336)

  1. Taking into account animal welfare and moving away from unnecessary animal research (the results presented in this article seem to be unnecessary from a scientific point of view), it is necessary to explain what the specific purpose of this research was? What do they contribute to the broadly understood literature on this topic? They did not examine anything except the effect of cholesterol, and this has been widely known in the literature for several decades. Additionally, most studies in the literature, in addition to the effects of dietary cholesterol supplementation on the body, also present the impact of other dietary supplements, which are atherogenic. Maybe if it were part of such research, i.e. research on some other factor in the diet involving cholesterol (hypercholesterolemic model), it would not be so blatant.

Response:   We appreciate the Reviewer for highlighting the important issue. So far, we have found limited studies investigating the impact cholesterol diet dos-age on plasma lipid profile or obesity. However, the experimental design of these studies involved multiple variables, such as the interactions between cholesterol diet and saturated fatty acids, (n-3) polyunsaturated fatty acids, fat, and fructose [19-21]. Therefore, it was unclear whether specific dosages of cholesterol diets can induce particular symptoms in hamsters. This study is the first to show that varying specific dosages of cholesterol diets can independently induce specific symptoms in hamsters, such as elevated LDL-C, VLDL-C, and hepatic lipid accumulation, without any additional treatment. These results are crucial for future studies using hamster models to evaluate the effects of drugs or natural substances on hyperlipidemia, as they provide important information on the appropriate cholesterol dosage in the diet. This can help reduce the number of experimental groups and animals used, adhering to the 3Rs principle in the further animal experiments. (line 325-336)

  1. The presented discussion of the obtained results must ABSOLUTELY be extended to include data commonly known in the literature on this issue. Because it is ABSOLUTELY NOT TRUE that the quote from the authors of this manuscript "So far, there is limited research on how different dietary cholesterol levels impact cholesterol metabolism and antioxidant enzyme activity of hamsters. It remains unclear which cholesterol dosage best mimics human lipid metabolism and the underlying mechanisms involved” [Lines 71 to 73].

Response:   We express our thanks to the Reviewer for noting the crucial issue. Although numerous studies have used high-cholesterol diets to induce hyperlipidemia in hamsters, we have not found any scientific findings from a single study that reveals the impact of only one variable, "different dietary cholesterol levels," on lipid metabolism and lipid peroxidation in hamsters. Moreover, it lacks credibility to speculate on the impact of different dietary cholesterol levels on lipid metabolism and lipid peroxidation based on data obtained from different studies. (line 115-120)

  1. There is a huge amount of data on this subject in the literature. They may not apply to hamsters, but they apply to almost all other laboratory animals. I am the author or co-author of several such works over the last 15-20 years!
  2. Therefore, It is ABSOLUTELY necessary to clarify the validity of these studies. Therefore, among others in the conclusions, the innovativeness and application possibilities of this research should be explained, emphasized and indicated. And it's not that the use of another laboratory animal model, the Syrian hamster, is new. This is nothing new considering the body's overall response to different concentrations of dietary cholesterol. This answer is either the same or very similar in various laboratory animal models.

Response:   We are grateful to the Reviewer for identifying the significant issue. Compared to other laboratory animals, the advantage and uniqueness of using hamster for hyperlipidemia model has been added, and the innovativeness and application possibilities of this research has been explained, emphasized and indicated in the revised manuscript. (line 104-125, 325-336)

  1. The references used are appropriate, but given the existing data in the literature, they are very sparse. Increasing the number of references would perhaps help better substantiate the relevance of the presented results in the context of the existing body of research and provide a more comprehensive review of the literature. It would also be beneficial to include more recent research and a wider range of sources to compare and contrast with current findings.

Response:   We thank the Reviewer for the valuable suggestion. Some relevant and recent studies have been cited in the revised manuscript. (Reference 6-8, 16-23)

Reviewer 2 Report

Comments and Suggestions for Authors

In this manuscript, the authors sought to identify the optimal dietary cholesterol content for use in the study of the commonly used Syrian Hamster model that would match aberrant cholesterol metabolism in humans. To accomplish this, they fed groups of Syrian hamsters diets varying only in cholesterol content ( 0, 0.1, 0.2, 0.5 and 1%) for 8 weeks. They then measure key indicators of cholesterol handling and metabolism (LDL, HDL, TG, etc.) at the tissue and systemic levels. They examine the effect of increased dietary cholesterol on markers of oxidative stress and liver health. They conclude by performing regression analysis to identify the dietary content of cholesterol that would best replicate clinical manifestations of impaired cholesterol homeostasis in humans. Generally, increasing dietary cholesterol increased circulating and liver content of cholesterol forms, while decreasing markers of oxidative stress in liver and increasing it in heart.

General Comments
The paper is generally well written and organized, and the dose response approach used in the study is a major strength towards determining optimal dietary concentrations of cholesterol. Experimental methods are well described and generally appropriate for the purposes of this study. Some clarifications are needed in the methods section. Generally, the novelty of the study is overstated, as several other groups have examined the dose response of dietary cholesterol in hamsters, examining similar endpoints

Introduction:

The authors provide a good review of the known associations between circulating cholesterol and factors they measure in this manuscript. However, there is a focus on the obese/metabolic syndrome phenotype, without explicit links to aberrant cholesterol metabolism in these conditions, outside of “dyslipidemia” used in line 27. I would recommend providing a stronger link between obesity and cholesterol homeostasis, if obesity is the major health condition of concern here.

29-30- Perhaps beyond the scope of this model study, the uncertainty surrounding dietary cholesterol’s role in lipoprotein metabolism is not well represented here. The implication in this statement is that dietary cholesterol (in humans) does not always translate to hypercholesterolemia, but the literature cited here is done in a mouse model. I would recommend a more nuanced justification of examining dietary cholesterol (Reviewed here: PMID: 38282001).

Line 59-61- It is unclear why this sentence is included in the introduction as no examination of glucagon sensitivity or factors related to glucose homeostasis is included in this paper. I recommend removing.

Line 71- The use of “limited research” as it pertains to cholesterol metabolism in hamsters and lipoprotein metabolism is overstated. A cursory search of the literature provides many cholesterol dose response studies in hamsters and other models (PMID: 19509184, PMID: 21143982, PMID: 19536872 to name a few). I would suggest removing this first sentence, and start with “It remains unclear which cholesterol dosage…”

Line 76- It is unclear the term “potential damage” refers to in this context. Outside of Liver enzymes, no measure as a result of cholesterol (atherosclerotic or otherwise) is measured. Please clarify.

Methods:

Generally, the methods are well described and comprehensive.

Two issues are noted

1. Regarding the ARRIVE guidelines, the sample size (2a) per group is listed as 10 (line 87). However, in multiple tables, the authors consistently list 8-10 animals per dietary group. This implies either exclusion occurred or not all groups were equal at the onset. The authors should rectify this either by including reasons for excluding animals from the data, or if they did not finish the dietary regimen. If animals did “drop out” this should be noted in the Animals and Experimental Diets section.

2. The description of the statistical analysis used for correlations is not sufficiently detailed. The rationale for model to be used is not described; it is unclear why broken line analysis was used, especially when the trend of the response to dietary cholesterol did not seem to vary depending on the dosage.

 It should be clarified why linear responses were favored over other models, especially when the corresponding points were far outside the better fitting model (eg. Figure 1B&D).

123- This is the first use of TBARS, and it has not been stated what TBARS is meant to measure. The acronym should be spelled out and the purpose of its use should be defined.

Results:

Results are generally reported well, and tables are clear.

It is recommended that for table 7 and 8 that abbreviations used in the table are included in the legend.

Section 3.4.

The descriptions of the data in this section require some clarity. It would be helpful to include a small statement at the beginning of this section referring to the goal of these analysis. Something to the effect of “Using the dose response data, we sought to determine the content at which circulating cholesterol levels match those values outlined in the Adult Treatment Panel III (ATP III).”

Since these are projections, language to imply the predictive nature of these data is warranted. E.g. line 228: “Below 0.13%, it is expected that plasma cholesterol would be below 200mg/dL. Further, it is expected that beyond 0.23% plasma cholesterol would exceed 240mg.”

This is especially relevant for line 231, as dietary supplementation did not go beyond 1%, so 1.23% is far outside the predictive range for these models.

It is unclear why “stabilization” is relevant in these models, as these points are not mentioned in the discussion. It is possible that at these concentrations, intestinal absorption and/or the liver’s capacity to produce lipoproteins has been reached, but again, this is only cursorily discussed.

Figure 1.

Generally- Text of axis labels and marks are far too small. Consider increasing the size of these fonts. The number of decimal places included in these models is excessive, as are the r-squared values. Consider limiting these to the tenths or hundredths places. The text for these are also too small. Please indicate what red arrows are meant to signify in the legend.

There is disagreement between the intersecting lines on panel A and C. Please redraw.

It is unclear what the red arrow in panel E is meant to signify. Remove or clarify.

The “max 0.85” does not provide much to this figure and is inconsistent with other graphs. Remove, or draw the intersecting lines.

Discussion

254- Again, there are many other papers examining dose responses of cholesterol. Consider changing the language to how this adds to the literature regarding cholesterol and lipid metabolism

263-270- These effects do not explain the change in body weight. No paper is cited that links dysregulated cholesterol metabolism and lipase activity, and increased circulating fatty acids would lead to adipose hypertrophy, not reductions. In the study cited, mouse body weights were not decreased on High cholesterol diets but increased (due to high fat feeding). This paragraph needs to be completely rethought and reviewed. This weight effect is also in direct contrast to their focus on obesity in the introduction, and should be noted.

286- This appears not to be the right reference, as it is a meta-analysis.

I would consider a discussion of the weaknesses of this model as they relate to human cholesterol homeostasis, as well as weaknesses of the study in general. The front line treatments for excessive cholesterol have long been statins. How does this dietary model interact with the major pharmaceutical or dietary approaches to lowering cholesterol. What other aspects in this model need to be confirmed? Atherosclerotic plaques? Foam cell generation? Perturbations in other aspects of liver function? Are the weight changes observed at the concentrations suggested (0.97, 0.43) of concern for more complex metabolic etiologies like MASH or diabetes?

Author Response

Reviewer 2:

In this manuscript, the authors sought to identify the optimal dietary cholesterol content for use in the study of the commonly used Syrian Hamster model that would match aberrant cholesterol metabolism in humans. To accomplish this, they fed groups of Syrian hamsters diets varying only in cholesterol content ( 0, 0.1, 0.2, 0.5 and 1%) for 8 weeks. They then measure key indicators of cholesterol handling and metabolism (LDL, HDL, TG, etc.) at the tissue and systemic levels. They examine the effect of increased dietary cholesterol on markers of oxidative stress and liver health. They conclude by performing regression analysis to identify the dietary content of cholesterol that would best replicate clinical manifestations of impaired cholesterol homeostasis in humans. Generally, increasing dietary cholesterol increased circulating and liver content of cholesterol forms, while decreasing markers of oxidative stress in liver and increasing it in heart.

General Comments
The paper is generally well written and organized, and the dose response approach used in the study is a major strength towards determining optimal dietary concentrations of cholesterol. Experimental methods are well described and generally appropriate for the purposes of this study. Some clarifications are needed in the methods section. Generally, the novelty of the study is overstated, as several other groups have examined the dose response of dietary cholesterol in hamsters, examining similar endpoints

 Response:   We thank the Reviewer for his/her valuable suggestions. Below are our point-by-point responses to the Reviewer's comments

Introduction:

The authors provide a good review of the known associations between circulating cholesterol and factors they measure in this manuscript. However, there is a focus on the obese/metabolic syndrome phenotype, without explicit links to aberrant cholesterol metabolism in these conditions, outside of “dyslipidemia” used in line 27. I would recommend providing a stronger link between obesity and cholesterol homeostasis, if obesity is the major health condition of concern here.

Response:   We thank the Reviewer for the suggestion. A link between obesity and cholesterol homeostasis was provided in the Introduction section of the revised manuscript. (line 40-46)

29-30- Perhaps beyond the scope of this model study, the uncertainty surrounding dietary cholesterol’s role in lipoprotein metabolism is not well represented here. The implication in this statement is that dietary cholesterol (in humans) does not always translate to hypercholesterolemia, but the literature cited here is done in a mouse model. I would recommend a more nuanced justification of examining dietary cholesterol (Reviewed here: PMID: 38282001).

Response:   We thank the Reviewer for the suggestion. The sentence was deleted in the revised manuscript.

Line 59-61- It is unclear why this sentence is included in the introduction as no examination of glucagon sensitivity or factors related to glucose homeostasis is included in this paper. I recommend removing.

Response:   We thank the Reviewer for the suggestion. The sentence was deleted in the revised manuscript.

Line 71- The use of “limited research” as it pertains to cholesterol metabolism in hamsters and lipoprotein metabolism is overstated. A cursory search of the literature provides many cholesterol dose response studies in hamsters and other models (PMID: 19509184, PMID: 21143982, PMID: 19536872 to name a few). I would suggest removing this first sentence, and start with “It remains unclear which cholesterol dosage…”

Response:   We appreciate the Reviewer for highlighting the important issue. So far, we have found limited studies investigating the impact cholesterol diet dosage on plasma lipid profile or obesity. However, the experimental design of these studies contained more than one variable, e.g. interaction between cholesterol diet and saturated fatty acid, (n-3) polyunsaturated fatty acid, fat and fructose [Am J Physiol Endocrinol Metab. 2009 Aug;297(2):E462-73.; Nutr Metab (Lond). 2010 Dec 10:7:89; Mol Nutr Food Res. 2009 Jul;53(7):921-30.]. Therefore, it was unclear whether specific dosages of cholesterol diets can induce particular symptoms in hamsters. This study is the first to demonstrate that different specific dosages of cholesterol diets can induce particular symptoms, such as elevated LDL-C, VLDL-C, and hepatic lipid accumulation, in hamsters without other treatment. These results are crucial for future studies using hamster models to evaluate the effects of drugs or natural substances on hyperlipidemia, as they provide important information on the appropriate cholesterol dosage in the diet. This can help reduce the number of experimental groups and animals used, adhering to the 3Rs principle in the further animal experiments. (line 325-336)

Line 76- It is unclear the term “potential damage” refers to in this context. Outside of Liver enzymes, no measure as a result of cholesterol (atherosclerotic or otherwise) is measured. Please clarify.

Response:   We express our gratitude to the Reviewer for the valuable suggestion. The term “potential damage” was deleted in the revised manuscript. The relationship between dietary cholesterol and atherosclerosis is an important and complex issue that requires further comprehensive investigation.

Methods:

Generally, the methods are well described and comprehensive.

Two issues are noted

  1. Regarding the ARRIVE guidelines, the sample size (2a) per group is listed as 10 (line 87). However, in multiple tables, the authors consistently list 8-10 animals per dietary group. This implies either exclusion occurred or not all groups were equal at the onset. The authors should rectify this either by including reasons for excluding animals from the data, or if they did not finish the dietary regimen. If animals did “drop out” this should be noted in the Animals and Experimental Diets section.

Response:   We are grateful to the Reviewer for identifying the significant issue. Hamsters that could not provide enough volume of plasma for all analyses were excluded. (line 148-149)

  1. The description of the statistical analysis used for correlations is not sufficiently detailed. The rationale for model to be used is not described; it is unclear why broken line analysis was used, especially when the trend of the response to dietary cholesterol did not seem to vary depending on the dosage.

 It should be clarified why linear responses were favored over other models, especially when the corresponding points were far outside the better fitting model (eg. Figure 1B&D).

Response:   Thank you to the Reviewer for bringing the important issue to our attention. In this study, we used broken line analysis to evaluate the relationship between dietary cholesterol dosage and response variables. This method helps identify threshold effects or changes in response patterns that may not be evident in a linear model. Our rationale was to ensure we didn't overlook significant non-linear relationships or thresholds, providing a more nuanced understanding of the data, particularly when the response might shift at certain dosage levels. (line 291-296)

123- This is the first use of TBARS, and it has not been stated what TBARS is meant to measure. The acronym should be spelled out and the purpose of its use should be defined.

Response:   We thank the Reviewer for the valuable suggestion. The acronym of TBARS was spelled out and the purpose of its use was defined in the revised manuscript. (line 182)

Results:

Results are generally reported well, and tables are clear.

It is recommended that for table 7 and 8 that abbreviations used in the table are included in the legend.

Response:   We thank the Reviewer for the valuable suggestion. Abbreviations used in the table 7 and 8 were included in the legend. (line 277, 287)

Section 3.4.

The descriptions of the data in this section require some clarity. It would be helpful to include a small statement at the beginning of this section referring to the goal of these analysis. Something to the effect of “Using the dose response data, we sought to determine the content at which circulating cholesterol levels match those values outlined in the Adult Treatment Panel III (ATP III).”

Response:   We thank the Reviewer for the valuable suggestion. The description of the data was added in the revised manuscript. (line 291-298)

Since these are projections, language to imply the predictive nature of these data is warranted. E.g. line 228: “Below 0.13%, it is expected that plasma cholesterol would be below 200mg/dL. Further, it is expected that beyond 0.23% plasma cholesterol would exceed 240mg.”

This is especially relevant for line 231, as dietary supplementation did not go beyond 1%, so 1.23% is far outside the predictive range for these models.

Response:   We thank the Reviewer for the valuable suggestion. Language to imply the predictive nature of these data was warranted in the revised manuscript. (line 298, 300, 312, 426)

It is unclear why “stabilization” is relevant in these models, as these points are not mentioned in the discussion. It is possible that at these concentrations, intestinal absorption and/or the liver’s capacity to produce lipoproteins has been reached, but again, this is only cursorily discussed.

Response:   We thank the reviewer for the valuable suggestion. The term "stabilization" has been corrected in the revised manuscript. (line 305)

Figure 1.

Generally- Text of axis labels and marks are far too small. Consider increasing the size of these fonts. The number of decimal places included in these models is excessive, as are the r-squared values. Consider limiting these to the tenths or hundredths places. The text for these are also too small. Please indicate what red arrows are meant to signify in the legend.

There is disagreement between the intersecting lines on panel A and C. Please redraw.

It is unclear what the red arrow in panel E is meant to signify. Remove or clarify.

The “max 0.85” does not provide much to this figure and is inconsistent with other graphs. Remove, or draw the intersecting lines.

Response:   We thank the reviewer for the valuable suggestion. Figure 1 was revised according to the Reviewer’s suggestion.

Discussion

254- Again, there are many other papers examining dose responses of cholesterol. Consider changing the language to how this adds to the literature regarding cholesterol and lipid metabolism

Response:   We express our thanks to the Reviewer for noting the crucial issue. Comparison between this study and other studies examining dose responses of cholesterol was added in the Discussion section of the revised manuscript. (line 325-336)

263-270- These effects do not explain the change in body weight. No paper is cited that links dysregulated cholesterol metabolism and lipase activity, and increased circulating fatty acids would lead to adipose hypertrophy, not reductions. In the study cited, mouse body weights were not decreased on High cholesterol diets but increased (due to high fat feeding). This paragraph needs to be completely rethought and reviewed. This weight effect is also in direct contrast to their focus on obesity in the introduction, and should be noted.

Response:   We appreciate your thorough review and ensure these points are clearly integrated into the revised manuscript. (line 340-347)

286- This appears not to be the right reference, as it is a meta-analysis.

Response:   We thank the reviewer for the valuable suggestion. The description was deleted in the revised manuscript.

I would consider a discussion of the weaknesses of this model as they relate to human cholesterol homeostasis, as well as weaknesses of the study in general. The front line treatments for excessive cholesterol have long been statins. How does this dietary model interact with the major pharmaceutical or dietary approaches to lowering cholesterol. What other aspects in this model need to be confirmed? Atherosclerotic plaques? Foam cell generation? Perturbations in other aspects of liver function? Are the weight changes observed at the concentrations suggested (0.97, 0.43) of concern for more complex metabolic etiologies like MASH or diabetes?

Response:   We thank the Reviewer for drawing attention to the important matter. So far, we have found limited studies investigating the impact cholesterol diet dosage on plasma lipid profile or obesity. However, the experimental design of these studies contained more than one variable, e.g. interaction between cholesterol diet and saturated fatty acid, (n-3) polyunsaturated fatty acid, fat and fructose [Am J Physiol Endocrinol Metab. 2009 Aug;297(2):E462-73.; Nutr Metab (Lond). 2010 Dec 10:7:89; Mol Nutr Food Res. 2009 Jul;53(7):921-30.]. Therefore, it was unclear whether specific dosages of cholesterol diets can induce particular symptoms in hamsters. This study is the first to demonstrate that different specific dosages of cholesterol diets can induce particular symptoms, such as elevated LDL-C, VLDL-C, and hepatic lipid accumulation, in hamsters without other treatment. These results are crucial for future studies using hamster models to evaluate the effects of drugs or natural substances on hyperlipidemia, as they provide important information on the appropriate cholesterol dosage in the diet. This can help reduce the number of experimental groups and animals used, adhering to the 3Rs principle in the further animal experiments. (line 325-336)

Round 2

Reviewer 1 Report

Comments and Suggestions for Authors

The article submitted for review concerns the influence of varied dietary cholesterol levels on lipid metabolism in hamsters. The manuscript is a full article examining the effects of different levels of dietary cholesterol on obesity, the plasma lipid profile, lipid accumulation, potential damage, lipid and metabolizing enzymes. The research was also aimed at determination of the optimal cholesterol dosage for dietary supplementation in hamster models providing recommendations for future research on obese or hyperlipidemic animal models.

The authors have introduced modifications, corrections and improvements in the current version of the manuscript. completed the missing data and results, extended the introduction and discussion of the results. The current version of the manuscript looks much better and, despite persistent doubts regarding the validity of the research undertaken, may be accepted for further stages of publication.